# Tracking Temporal Dynamics of Vector Sets with Gaussian Process

## Abstract

Understanding the temporal evolution of sets of vectors is a fundamental challenge across various domains, including ecology, crime analysis, and linguistics. For instance, ecosystem structures evolve due to interactions among plants, herbivores, and carnivores; the spatial distribution of crimes shifts in response to societal changes; and word embedding vectors reflect cultural and semantic trends over time. However, analyzing such time-varying sets of vectors is challenging due to their complicated structures, which also evolve over time. In this work, we propose a novel method for modeling the distribution underlying each set of vectors using infinite-dimensional Gaussian processes. By approximating the latent function in the Gaussian process with Random Fourier Features, we obtain compact and comparable vector representations over time. This enables us to track and visualize temporal transitions of vector sets in a low-dimensional space. We apply our method to both sociological data (crime distributions) and linguistic data (word embeddings), demonstrating its effectiveness in capturing temporal dynamics. Our results show that the proposed approach provides interpretable and robust representations, offering a powerful framework for analyzing structural changes in temporally indexed vector sets across diverse domains.

## 1 Introduction

Analyzing dynamically evolving vector sets is a critical challenge across various domains. Here, each vector typically represents a data point characterized by multiple attributes (e.g., spatial coordinates, statistical properties, or semantic features), and the entire set encodes the distribution of such points at a given time. For example, in ecology, the interaction among plants, herbivores, and carnivores leads to temporal changes in habitats (Odum, 1971), and analyzing these temporal transitions (Kass et al., 2018) can help reveal ecosystem dynamics. In sociology, social phenomena such as population movements and crime incidents can be studied over time across specific regions (Murakami & Yamagata, 2019; Murakami et al., 2020), enabling an understanding of changing patterns. In linguistics, understanding how the meaning of a particular word changes over time (Kulkarni et al., 2015; Hamilton et al., 2016; Aida et al., 2021) can be aided by acquiring embeddings from usage examples and analyzing the resulting collections (Aida & Bollegala, 2023; Nagata et al., 2023). Thus, understanding the *temporal transitions of spatially distributed sets of vectors* is foundational for revealing the underlying structures and interactions of such phenomena.

Previous studies have proposed various methods to estimate vector sets at particular time periods (e.g., species distributions, crime locations, word usage embeddings) (Kass et al., 2018; Murakami & Yamagata, 2019; Murakami et al., 2020; Weisburd et al., 2006; Hachadoorian et al., 2011; Aida & Bollegala, 2023). However, while these approaches are effective at modeling states at specific times, they are limited in their ability to reveal the *mechanisms of temporal variation across the entire vector set* and the *relationships between different sets of vectors*.

To address these limitations, we propose a novel method that represents each vector set at a given time as a single distribution using a compact approximation of *Gaussian processes* (GPs) via *Random Fourier Features* (RFF) (Rahimi & Recht, 2007). RFF approximates a distribution using a combination of $K$ cosine functions, allowing complicated distributions to be represented as $K$-dimensional real vectors in frequency space.

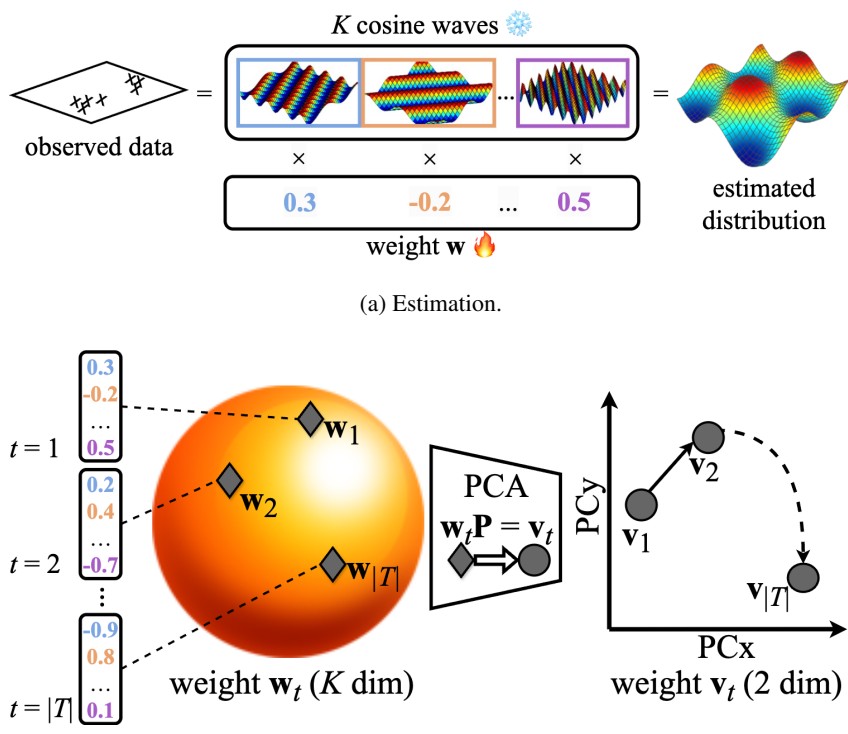

(a) Estimation.

(b) Tracking.

Figure 1: Overview. (a) We randomly sample $K$ cosine basis functions and estimate their corresponding weights from the observed vector set $\mathbb{R}^{N \times 2}$ at each time step $t \in T$. This allows us to represent the temporal dynamics of the vector set as a $K$-dimensional vector $\mathbf{w}_t \in \mathbb{R}^K$, for each time step. (b) After that, we apply Principal Component Analysis (PCA) to the set of weight vectors $\{\mathbf{w}_t\}_{t=1}^{|T|}$ to obtain two-dimensional projections $\mathbf{v}_t \in \mathbb{R}^2$. This enables compact visualization and analysis of temporal transitions of vector sets over time.

This compact representation is derived from GPs, which can be viewed as an infinite-dimensional generalization of Gaussian distributions over function spaces. GPs are known for their flexibility in capturing complicated, nonparametric structures, making them well-suited for modeling the temporal changes of vector sets. However, traditional GPs retain information about all data points to model complicated distributions, making it challenging to extract essential distributional characteristics. By leveraging RFF, we retain the expressive power of GPs while enabling a tractable and interpretable analysis of transitions of vector sets over time. We evaluate the effectiveness of the proposed method by applying it to spatial crime data and analyzing semantic changes in words via sets of word usage vectors.

## 2 TRACKING VECTOR SET TRANSITIONS VIA RFF

To analyze how sets of vectors evolve over time, we need a way to compactly and meaningfully represent each set at a given time. Rather than treating vectors individually, we represent each set as a probability density function over the vector space, capturing global structures. This enables us to move beyond pointwise statistics or aggregate moments and to treat the entire set as a unified object with rich geometric and statistical properties. To achieve this, we model the density of vectors on a target space $\mathcal{X}$ using the *Gaussian Process Density Sampler* (GPDS) (Murray et al., 2008):

$$p(\mathbf{x}) = \frac{\sigma(f(\mathbf{x}))}{Z}, \quad Z = \int_{\mathcal{X}} \sigma(f(\mathbf{x})) \mathrm{d}\mathbf{x}. \tag{1}$$

Here, $\sigma(x) = 1/(1 + e^{-x})$ is the sigmoid function, $f$ is a sample from a Gaussian process $\mathrm{GP}(0, k(\mathbf{x}, \mathbf{x}'))$, where $k$ is a kernel function that defines similarity between vectors in space $\mathcal{X}$.

The function $f$ serves as the main learnable component, defining the unnormalized log-density over the space. However, the density equation 1 is intractable due to the normalization constant $Z$, which involves integration over the entire space $\mathcal{X}$. To circumvent this issue, we apply contrastive learning (Gutmann & Hyvärinen, 2012), treating $Z$ as a latent variable to be learned by maximizing the following discriminative objective:

$$\sum_{\mathbf{x} \in \mathcal{D}} \log p(z{=}1 \,|\, \mathbf{x}) + \sum_{\mathbf{x}' \in \overline{\mathcal{D}}} \log p(z{=}0 \,|\, \mathbf{x}'), \tag{2}$$

where $\mathcal{D}$ is the set of positive, observed samples and $\overline{\mathcal{D}}$ is a set of randomly sampled negative examples and binary variable $z$ indicates whether data is generated from the target distribution or random distribution. Conventionally, the function $f$ is represented non-parametrically as a vector of function values at $N$ data points $\mathbf{f} = (f(\mathbf{x}_1), \cdots, f(\mathbf{x}_N))^\mathsf{T}$. To model the density function $f$ non-parametrically from the data, GP-based approach usually requires computing the kernel matrix between $N$ points, which incurs $O(N^3)$ complexity and lacks explicit global information.

Instead of explicitly computing the kernel function $k(\mathbf{x}, \mathbf{x}')$, which corresponds to an inner product in an infinite-dimensional feature space, we approximate it as a dot product in a randomized $K$-dimensional space $k(\mathbf{x}, \mathbf{x}') \approx \phi(\mathbf{x})^\mathsf{T}\phi(\mathbf{x}')$. Here, $\phi(\mathbf{x})$ is a finite-dimensional random feature map constructed via Fourier transform and Monte Carlo integration, defined as: $\phi(\mathbf{x}) = (\phi_1(\mathbf{x}), \phi_2(\mathbf{x}), \ldots, \phi_K(\mathbf{x}))^\mathsf{T}$. For the Gaussian kernel, the feature function can be expressed as

$$\begin{cases} \phi_k(\mathbf{x}) = \sqrt{\dfrac{2}{K}} \cos(\boldsymbol{\omega}_k^\mathsf{T}\mathbf{x} + b_k), \\ \boldsymbol{\omega}_k \sim \mathcal{N}(\mathbf{0}, \sigma^2\mathbf{I}), \; b_k \sim \mathrm{Unif}[0,1]. \end{cases}$$

This is known as *Random Fourier Features* (RFF) (Rahimi & Recht, 2007). Each $\boldsymbol{\omega}_k$ is a frequency vector and $b_k$ is a phase offset. Under this representation, a sample $f$ from the GP can then be expressed as a linear model $\mathbf{f} = \boldsymbol{\Phi}\mathbf{w}$:

$$\underbrace{\begin{pmatrix} f(\mathbf{x}_1) \\ f(\mathbf{x}_2) \\ \vdots \\ f(\mathbf{x}_N) \end{pmatrix}}_{\mathbf{f}} = \underbrace{\begin{pmatrix} \phi(\mathbf{x}_1)^\mathsf{T} \\ \phi(\mathbf{x}_2)^\mathsf{T} \\ \vdots \\ \phi(\mathbf{x}_N)^\mathsf{T} \end{pmatrix}}_{\boldsymbol{\Phi}} \underbrace{\begin{pmatrix} w_1 \\ \vdots \\ w_K \end{pmatrix}}_{\mathbf{w}} \tag{3}$$

where $\boldsymbol{\Phi}$ is a design matrix composed of feature vectors $[\phi(\mathbf{x}_1)^\mathsf{T}; \cdots; \phi(\mathbf{x}_N)^\mathsf{T}]$, and $\mathbf{w} \in \mathbb{R}^K$ is a weight vector. This formulation compactly represents a complicated vector distribution via a single $K$-dimensional vector. Although Equation 3 is written for the training instances $\mathbf{x}_i$, the same relation can be used to approximate $f(\mathbf{x})$ for any input $\mathbf{x}$ once $\mathbf{w}$ is obtained. Our focus is not on generalization, but on modeling and comparing the weight vectors to analyze temporal dynamics.

In our framework, we fix the sampled feature functions $\phi_k(\cdot)$ and estimate the optimal weights $\mathbf{w}$ using a random-walk Metropolis-Hastings algorithm (Bishop & Nasrabadi, 2006). We apply the same feature functions across all time steps to obtain comparable weight vectors $\mathbf{w}_t$ for each time $t \in \{1, 2, \ldots, |T|\}$. Finally, we analyze the set of weight vectors $\{\mathbf{w}_1, \ldots, \mathbf{w}_{|T|}\}$. Since they reside in a $K$-dimensional space and are not directly interpretable, we apply PCA to reduce the dimensionality and visualize the temporal transitions of vector sets in a compact 2D space (Figure 1).

## 3 EXPERIMENTS

We evaluate our proposed method on both synthetic and real-world data. The real-world evaluations are conducted on two domains: spatial distributions of crime incidents in Chicago, and semantic changes in English words over two historical periods. Our goal is to demonstrate that the temporal dynamics of diverse vector sets can be effectively captured and interpreted via our method.

### 3.1 PRELIMINARY INVESTIGATIONS ON SYNTHETIC DATA

Before evaluating our method on real-world data, we conducted experiments using synthetic 2-dimensional vector sets to assess its ability to capture known distributional transitions over time.

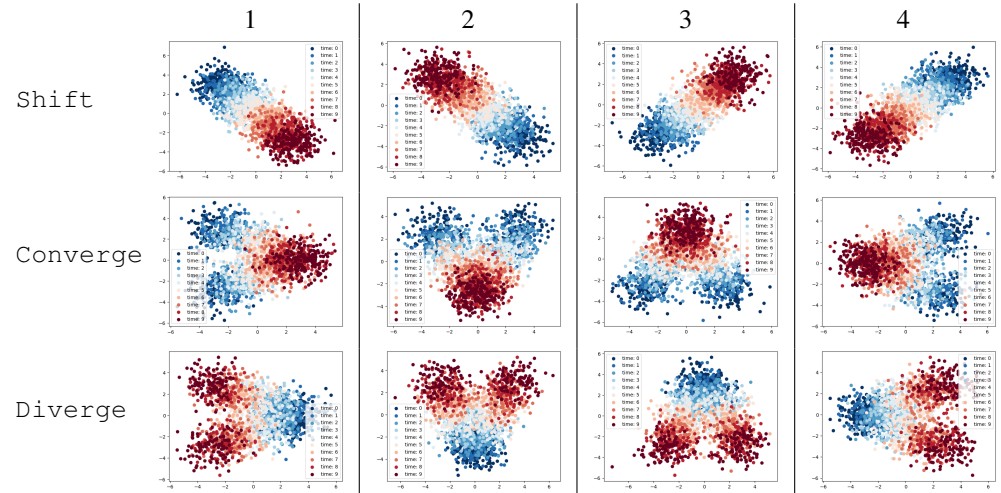

Figure 2: Synthetic datasets across 10 time steps, transitioning from blue (earliest) to red (latest). Each row represents a movement type of point set movement: Shift, Converge, and Diverge. Each column represents a different instance within that type (1 to 4). For example, the top-left panel corresponds to the first instance of the Shift movement type (denoted as Shift_1).

These synthetic datasets were designed to simulate representative movement patterns of vector sets, including linear shifts (Shift), merging (Converge), and splitting (Diverge).

### 3.1.1 SETTINGS

**Dataset Generation**  We generated 12 synthetic datasets in total, each of which we refer to as an *instance*. There are three movement types (Shift, Converge, Diverge) × four instances per type. Each dataset (e.g. Shift_1) consists of 10 time steps, each containing 200 points. In Converge and Diverge movement types, each vector set is composed of two subgroups of 100 points between merging or splitting components. Figure 2 illustrates these datasets, where vectors transition from blue (earliest) to red (latest) across time.

**Implementation Details**  We applied our method with $K = 30$ cosine basis functions to each vector set at each time step. This resulted in 120 inferred RFF weight vectors (3 movement types × 4 instances × 10 time steps). To visualize the temporal dynamics, we performed PCA jointly on all 120 weight vectors and plotted the trajectories in the principal component space.

### 3.1.2 RESULTS

We focus on two contrasting pairs of instances: Shift_1 vs. Shift_2, and Converge_4 vs. Diverge_4, which are visualized in Figure 3. In the Shift instances, a single Gaussian cluster move in opposite directions across time periods. The projected trajectories of the inferred RFF weights in the PCA space exhibit clear and symmetric patterns: the trajectory for Shift_1 curves to the left, while that of Shift_2 mirrors it to the right. This indicates that the temporal evolution of vector set distributions is captured as directional movement in the embedding space. Moreover, the ordering of the color-coded dots (from blue to red) aligns smoothly along the paths, indicating that our representation maintains temporal consistency across time steps.[1]

In the Converge and Diverge instances, the clusters undergo non-linear transformations involving either merging into a single mode (Converge) or splitting into two modes (Diverge). Despite these more complicated dynamics, our method successfully captures the divergence and convergence of distributions. In Converge_4, the PCA trajectory starts with two distinct clusters that gradually converge into one, resulting in a smooth inward-curving path. Conversely, Diverge_4 begins with a single cluster that bifurcates, producing an outward-diverging trajectory. Importantly, these

---

[1]While the heatmaps surrounding each point reflect the general distribution of the vector sets, we observe that some estimated densities exhibit high values in regions devoid of actual data. Addressing this under/overestimation remains an open challenge for future work.

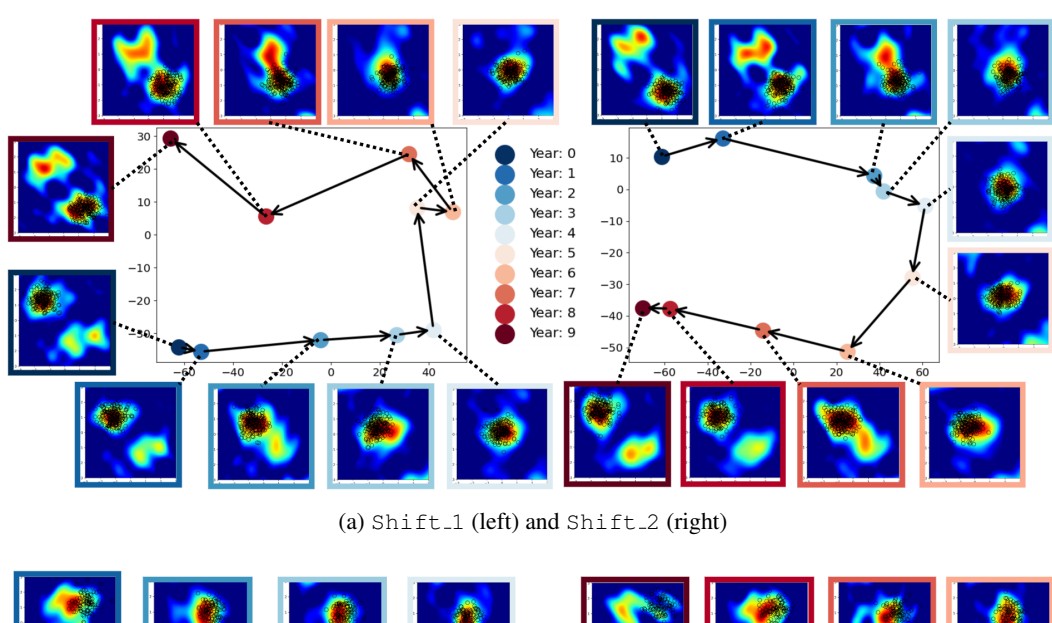

(a) `Shift_1` (left) and `Shift_2` (right)

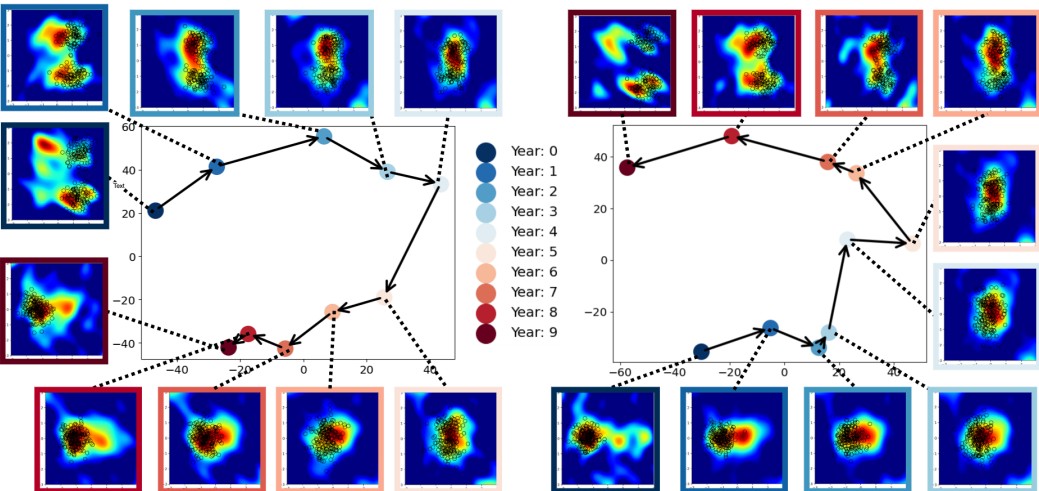

(b) `Converge_4` (left) and `Diverge_4` (right)

Figure 3: Trajectories of synthetic datasets in the top two principal components (PC1 and PC2), visualized for selected instances. We focus on two pairs of datasets with contrasting dynamics: `Shift_1` vs. `Shift_2`, and `Converge_4` vs. `Diverge_4`. The central plots show the projection of inferred 30-dimensional weights onto the PC1–PC2 space. Surrounding heatmaps represent the estimated distributions over the vector sets (represented by ○) at each time step.

temporal transitions are visible not only in how the PCA trajectories move (e.g., curving inward or outward), but also in the corresponding heatmaps: the number of high-density regions changes over time. Similar trends were observed in other instances as well, supporting the robustness of our method (see Appendix A for full results).

These results demonstrate that our approach provides a faithful and compact representation of temporal dynamics for vector sets. It robustly encodes both simple translational shifts and more nuanced topological changes such as the emergence or disappearance of density peaks. The trajectories in the low-dimensional space offer a useful summary for downstream analysis, such as clustering, anomaly detection, or change point identification.

## 3.2 CASE 1: CHICAGO CRIMES IN GEOGRAPHIC SPACE

Having confirmed in the previous section that our method accurately captures known distributional shifts in synthetic vector sets, we now turn to real-world data to further validate its practical utility.

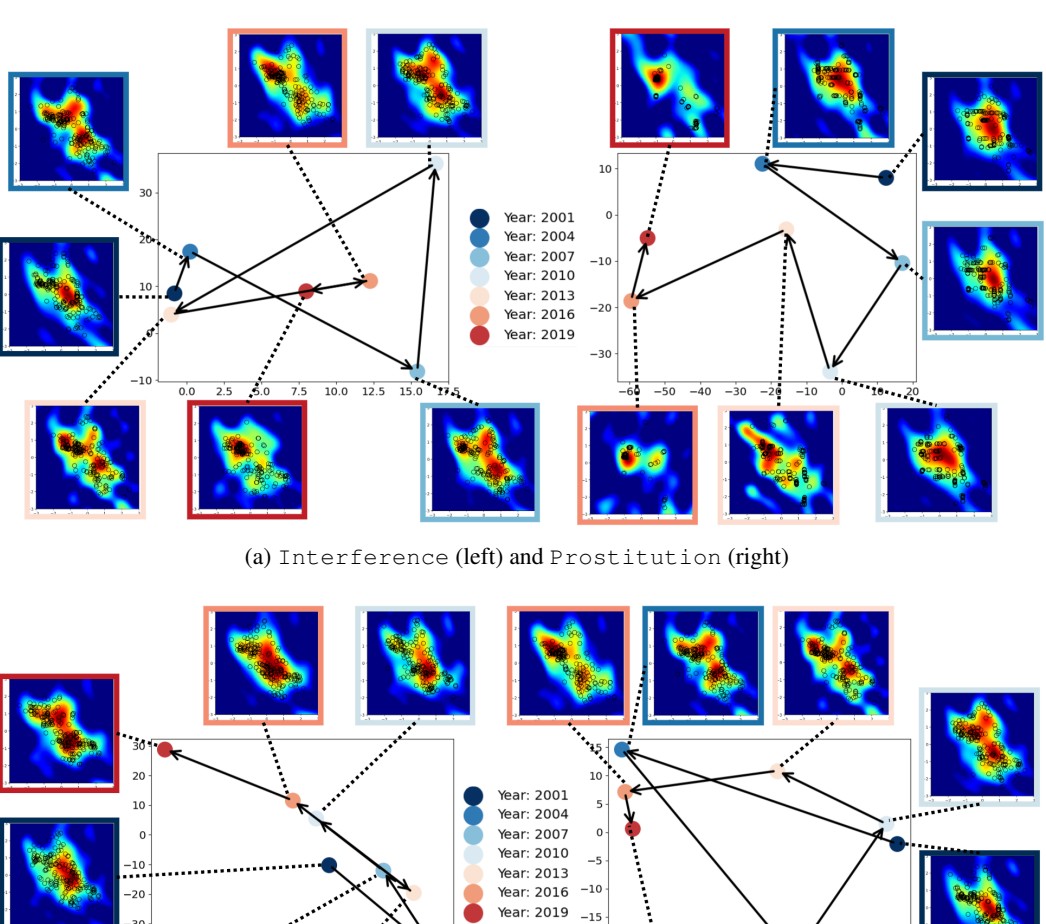

(a) `Interference` (left) and `Prostitution` (right)

(b) `Weapons` (left) and `Narcotics` (right)

Figure 4: Trajectories of Chicago Crimes datasets in the top two principal components (PC1 and PC2). The central plots show the projection of inferred 30-dimensional weights onto the PC1–PC2 space. Surrounding heatmaps represent the estimated distributions over the vector sets (represented by ○) at each time period.

As a first case study, we focus on spatio-temporal data: specifically, the geographic distribution of crime incidents in the city of Chicago. This dataset presents complicated real-world dynamics in both spatial and temporal dimensions, making it a suitable testbed for evaluating the effectiveness of our representation and tracking framework.

### 3.2.1 SETTINGS

We apply our method to the spatio-temporal distribution of crime incidents in Chicago. The dataset consists of crime reports recorded between 2001 and 2024 and includes the time of occurrence, location (latitude and longitude), and categorized crime types.[2] For the purpose of this study, we select four crime categories that exhibit distinct temporal trends: two with increasing incident counts – Interference with Public Officers (`Interference`) and Weapons Violations (`Weapons`)-and two with decreasing trends—Prostitution (`Prostitution`) and Narcotics (`Narcotics`).[3]

---

[2]https://data.cityofchicago.org

[3]See Figure 8 in Appendix B for spatial and temporal distributions.

For each crime type, we uniformly sample 200 incidents per year at seven time periods, spanning from 2001 to 2019 at three-year intervals (i.e., 2001, 2004, ..., 2019), resulting in a total of 1,400 samples per category. Spatial coordinates are centered and normalized. To model temporal changes, we estimate RFF weights $\mathbf{w}_t \in \mathbb{R}^K$ for each time period using a cosine basis matrix $\mathbf{\Phi}$ with $K = 30$ components. These weights represent the underlying spatial distribution of events.

### 3.2.2 RESULTS

Figure 4 illustrates the projection of estimated weights onto the top two principal components, along with the corresponding estimated heatmaps for each year. We focus on two representative pairs of crime categories: `Interference` vs. `Prostitution`, and `Weapons` vs. `Narcotics`.

For the first pair, both trajectories shift leftward along PC1 as time progresses. Notably, the `Prostitution` category moves farther to the left (toward $-50$), while the `Interference` category exhibits a milder shift and stabilizes around 10. The direction of movement is consistent across both trends but differs in magnitude. These changes align with the evolution of spatial distributions. As weights move leftward, the corresponding heatmaps become more peaked and localized, indicating increasing concentration of crime incidents in specific regions. This suggests that PC1 captures the degree of spatial concentration. A similar observation holds for the second pair. While both categories evolve over time, `Narcotics` shows a stronger trajectory along PC1, consistent with its sharper decline in incident frequency and localization. In contrast, the `Weapons` category remains more spatially dispersed over the years.

In addition to PC1, we observe a common trend along PC2 across all crime categories. As the PC2 coordinate increases, the corresponding heatmaps become more spatially diffuse and widespread, indicating a broadening of incident locations (`Interference` and `Narcotics`). Conversely, movement toward negative PC2 is associated with a contraction of the density into a more centralized region (`Prostitution` and `Weapons`). This suggests that PC2 captures the extent of spatial spread or dispersion. Taken together, these results demonstrate that our method effectively captures meaningful temporal dynamics in spatial vector patterns and generalizes from synthetic data to real-world spatio-temporal event streams.

### 3.3 CASE 2: LEXICAL SEMANTIC CHANGE IN VECTOR SPACE

Having verified the effectiveness of our method in tracking spatial vector transitions, we next turn to a more abstract domain: lexical semantics in natural language processing. Specifically, we apply our framework to the task of tracking lexical semantic change based on temporal transitions of contextual word embeddings.

### 3.3.1 SETTINGS

We use the English subtask of the SemEval-2020 Task 1 dataset (Schlechtweg et al., 2020), which provides labeled words from two time periods—specifically, the early period (1810–1860) and the late period (1960–2010). Each target word is associated with a set of usage examples for each period. To analyze the types of semantic shifts through the transitions of vector sets, we restrict our analysis to the words that are labeled as having undergone semantic change. We extract contextual word embeddings using a fine-tuned XLM-R large model (Cassotti et al., 2023), the top-performing system in the shared task. Each embedding is 1,024-dimensional.

To facilitate analysis, we reduce each embedding to a 2-dimensional vector using PCA. While our framework supports higher-dimensional representations, we adopt a 2-dimensional projection for interpretability, following prior work (Aida & Bollegala, 2025). These vector sets are treated as temporal instances, and we estimate RFF weights using $K = 30$ cosine basis functions. The inferred weights for all words and both periods are then jointly projected onto a 2-dimensional PCA space for visualization and comparison.

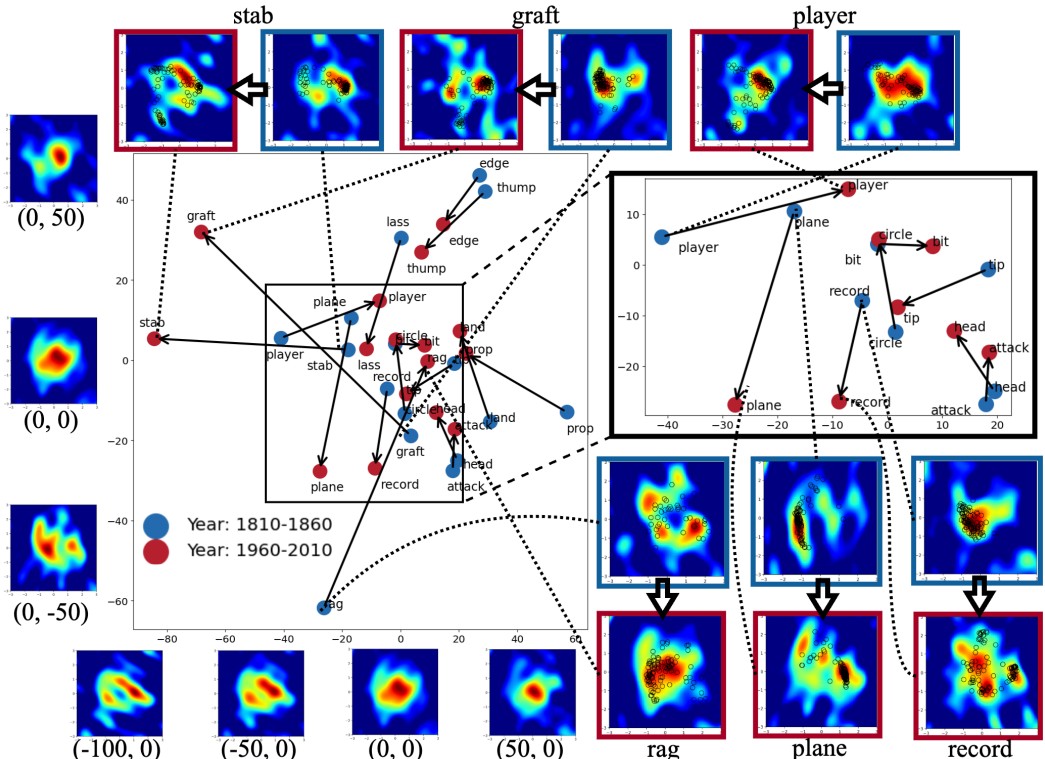

Figure 5: Trajectories of the SemEval dataset in the PC3 and PC5. The central plots show the projection of inferred 30-dimensional weights onto the PC3–PC5 space. To aid interpretation, each principal component axis is annotated with heatmaps that visualize the estimated distributions reconstructed by inverse-mapping a set of evenly spaced points (every 50 units) back to the original RFF space. Surrounding heatmaps represent the estimated distributions over the vector sets (represented by ○) at each time period.

### 3.3.2 RESULTS

Figure 5 illustrates the projection results in the PCA space for representative words.[4] We observe that words like stab and graft move strongly in the negative direction of PC3, corresponding to semantic broadening—from concrete meanings such as *to pierce* and *to graft plants*, to more abstract or figurative senses like *to criticize*, *medical transplant*, and *bribery*. In contrast, words like plane and record show dispersion along PC5. This expansion reflects the emergence of new technological senses (*airplane* and *audio record*), resulting in a broader and more dispersed distribution of contextual embeddings in the late period.

Interestingly, words such as player and rag also exhibit significant semantic shifts, yet along unexpected trajectories in the PCA space: Player shifts along PC3 instead of the expected PC5 (*actor* to *media player*), while rag moves along PC5 rather than PC3 (*old clothes* to *tabloid newspaper*). These different directions suggest that metaphorical extension and technological innovation manifest differently in the latent RFF representation space. Overall, these findings demonstrate that our method captures subtle variations in semantic change and offers an interpretable framework for analyzing such transitions via spatial and density-based cues.

## 4 RELATED WORK

Our work relates to two key research areas: (1) time-varying vector set modeling, which concerns tracking or representing dynamic distributions of unordered points, and (2) representation learning

---

[4]Note that PC1 and PC2 were excluded from the visualization, as they predominantly captured global trends not directly relevant to the semantic transitions of interest (see Figure 9 in Appendix C for details).

for distributional shift, particularly approaches that capture how data distributions evolve in latent space over time or across conditions.

## 4.1 TIME-VARYING VECTOR SET MODELING

This line of work deals with tracking and modeling entire sets of vectors as they evolve over time. Across fields such as linguistics, ecology, and sociology, it is common to first estimate spatially distributed vector sets (e.g., ecological distributions (Kass et al., 2018), crime locations (Murakami et al., 2020), word embeddings (Aida & Bollegala, 2023)) and then study their temporal dynamics. However, most of these methods focus on static snapshots or compare aggregate statistics between time slices, lacking the ability to model continuous transitions or interaction mechanisms among vector sets.

In point cloud research, related techniques have been developed to track time-varying point sets, particularly in 3D vision and physics simulations. For example, CloudLSTM (Zhang et al., 2020) and PSTNet (Fan et al., 2021) learn spatio-temporal dynamics in point sequences, while TPU-GAN (Li et al., 2022) captures motion-consistent upsampling of point clouds. Such methods emphasize spatial structure and temporal coherence but are often computationally intensive or tailored to 3D tasks. More broadly, Kalman filtering (Ding et al., 2024) and partial Wasserstein matching (Wang et al., 2022) have also been used for registration or tracking of evolving point sets. However, these methods assume stable point correspondences and are thus less suitable for analyzing unlabeled, distributionally shifting point sets such as those in social data or vector sets of language.

## 4.2 REPRESENTATION LEARNING FOR DISTRIBUTIONAL SHIFT

Dimensionality reduction methods such as PCA and t-SNE (van der Maaten & Hinton, 2008) have been widely used to embed vector sets into low-dimensional spaces for visualization. However, they primarily focus on preserving geometric or local neighborhood structures in static data and do not account for how distributions evolve. GP-LVM (Titsias & Lawrence, 2010) provides a probabilistic framework for learning latent manifolds, and its dynamic extension allows for temporal modeling. UMAP (McInnes et al., 2020) improves manifold learning with better preservation of global structure and density, but still lacks an explicit temporal component.

Recent advances have proposed explicit models for latent dynamics. For instance, DVBF (Karl et al., 2017) combines variational inference with dynamic systems to learn latent state transitions from high-dimensional observations. Time-lagged information bottleneck (Federici et al., 2024) encodes features that preserve predictive information at coarse time scales, while InterLatent (Li et al., 2025) models intermittent activation of latent factors over time. These methods excel at modeling dynamics at the level of individual sequences or samples, but they do not directly address how entire vector sets evolve as distributions over time. In contrast, we propose to model each vector set as a unified distribution via Random Fourier Features (Rahimi & Recht, 2007), enabling compact and interpretable tracking of global distributional transitions across time.

## 5 CONCLUSION

In this work, we proposed a novel method for modeling temporal transitions of vector sets by representing each set as a distribution approximated with Random Fourier Features (RFF). This compact representation enables interpretable analysis of set-level dynamics over time, going beyond existing methods that rely on static snapshots or per-sample states. Through experiments on spatial crime data and semantic change in vector space, we demonstrated that our method effectively captures meaningful distributional transitions in diverse real-world scenarios.

## LLM USAGE

We used a large language model solely for language editing and minor grammar improvements. All experimental design, implementation, and writing decisions were made by the authors.

ETHICS STATEMENT

This work does not involve human subjects, personally identifiable information, or sensitive data. All experiments are conducted using either publicly available datasets or synthetically generated data. We note the following considerations:

- The crime data used in our experiments captures aggregated statistics on when and where incidents occurred. Our analysis is restricted to spatio-temporal trends and does not attempt to infer attributes of individuals or communities.

- To the best of our knowledge, no ethical issues have been reported for the language data we used in our experiments. However, pretrained models may implicitly reflect societal biases Basta et al. (2019). While our focus is on analyzing distributional dynamics over time, we acknowledge the risk of amplifying or misinterpreting such biases through representation-based methods.

As our approach aims to capture evolving patterns in vector representations, it is crucial to carefully consider the possibility that statistical methods may unintentionally reinforce or obscure structural biases. We encourage further investigation into the ethical implications of modeling temporal dynamics in social data.

REPRODUCIBILITY STATEMENT

We will release the complete implementation as anonymized supplementary materials. All experiments were conducted using publicly available datasets and a publicly released embedding model. Further details of the data preparation, embedding procedure, and experimental settings are provided in section 3.

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

## A FULL RESULTS IN SUBSECTION 3.1

## B CHICAGO CRIME DATA IN SUBSECTION 3.2

## C FULL RESULTS IN SUBSECTION 3.3

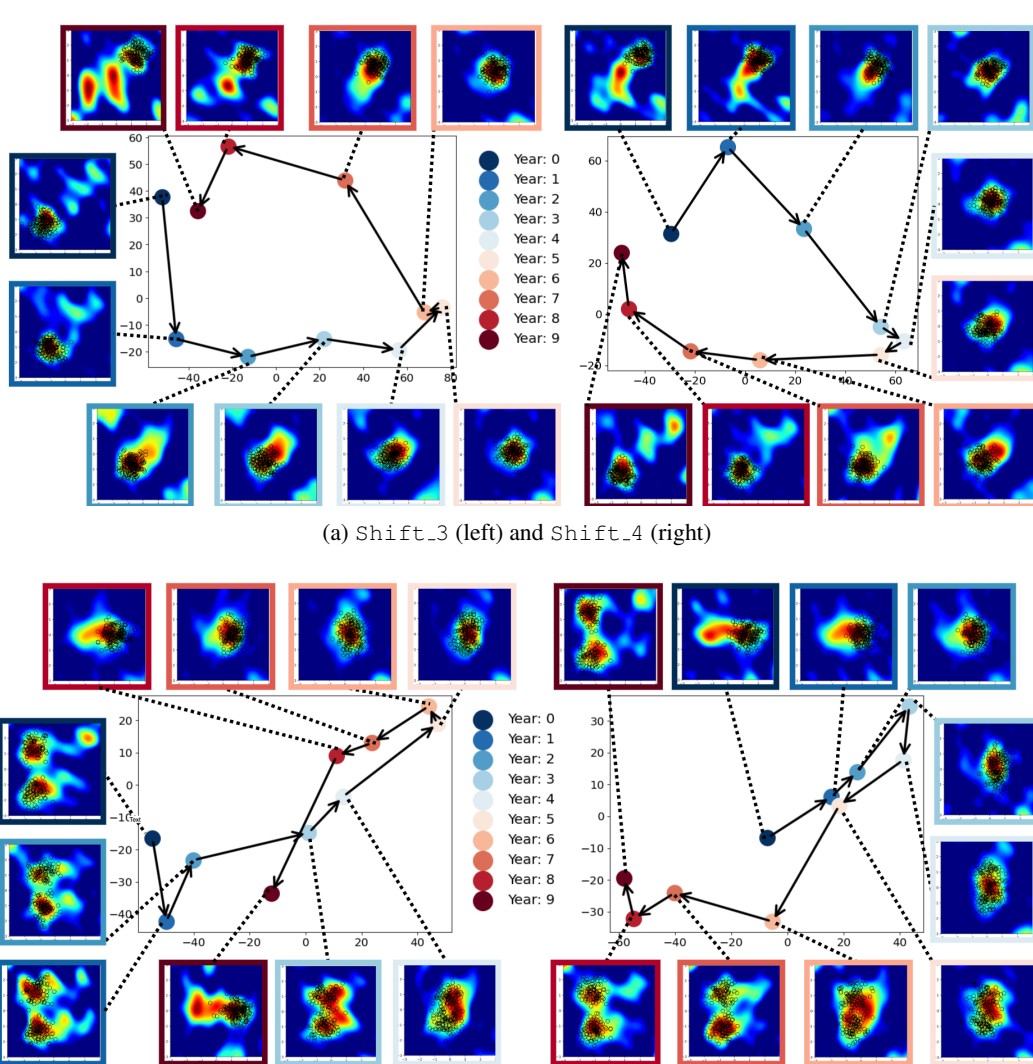

(a) Shift_3 (left) and Shift_4 (right)

(b) Converge_1 (left) and Diverge_1 (right)

Figure 6: Trajectories of synthetic datasets in the top two principal components (PC1 and PC2), visualized for selected instances. We focus on two pairs of datasets with contrasting dynamics: Shift_3 vs. Shift_4, and Converge_1 vs. Diverge_1. The central plots show the projection of inferred 30-dimensional weights ($K = 30$ Cosine waves) onto the PC1–PC2 space. Surrounding heatmaps represent the estimated distributions over the vector sets (represented by $\circ$) at each time step. These results illustrate that our method can effectively capture symmetric differences in the learned transition patterns across time.

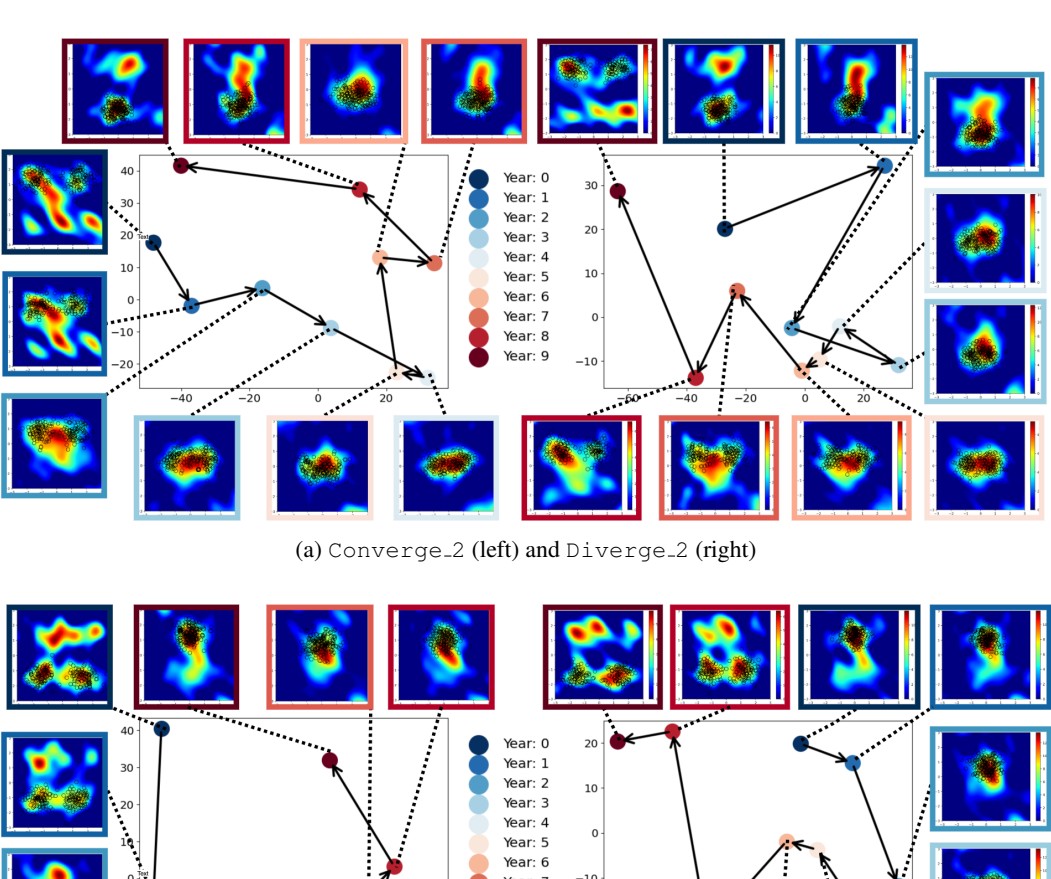

(a) `Converge_2` (left) and `Diverge_2` (right)

(b) `Converge_3` (left) and `Diverge_3` (right)

Figure 7: Trajectories of synthetic datasets in the top two principal components (PC1 and PC2), visualized for selected instances. We focus on two pairs of datasets with contrasting dynamics: `Converge_2` vs. `Diverge_2`, and `Converge_3` vs. `Diverge_3`. The central plots show the projection of inferred 30-dimensional weights ($K = 30$ Cosine waves) onto the PC1–PC2 space. Surrounding heatmaps represent the estimated distributions over the vector sets (represented by ∘) at each time step. These results illustrate that our method can effectively capture symmetric differences in the learned transition patterns across time.

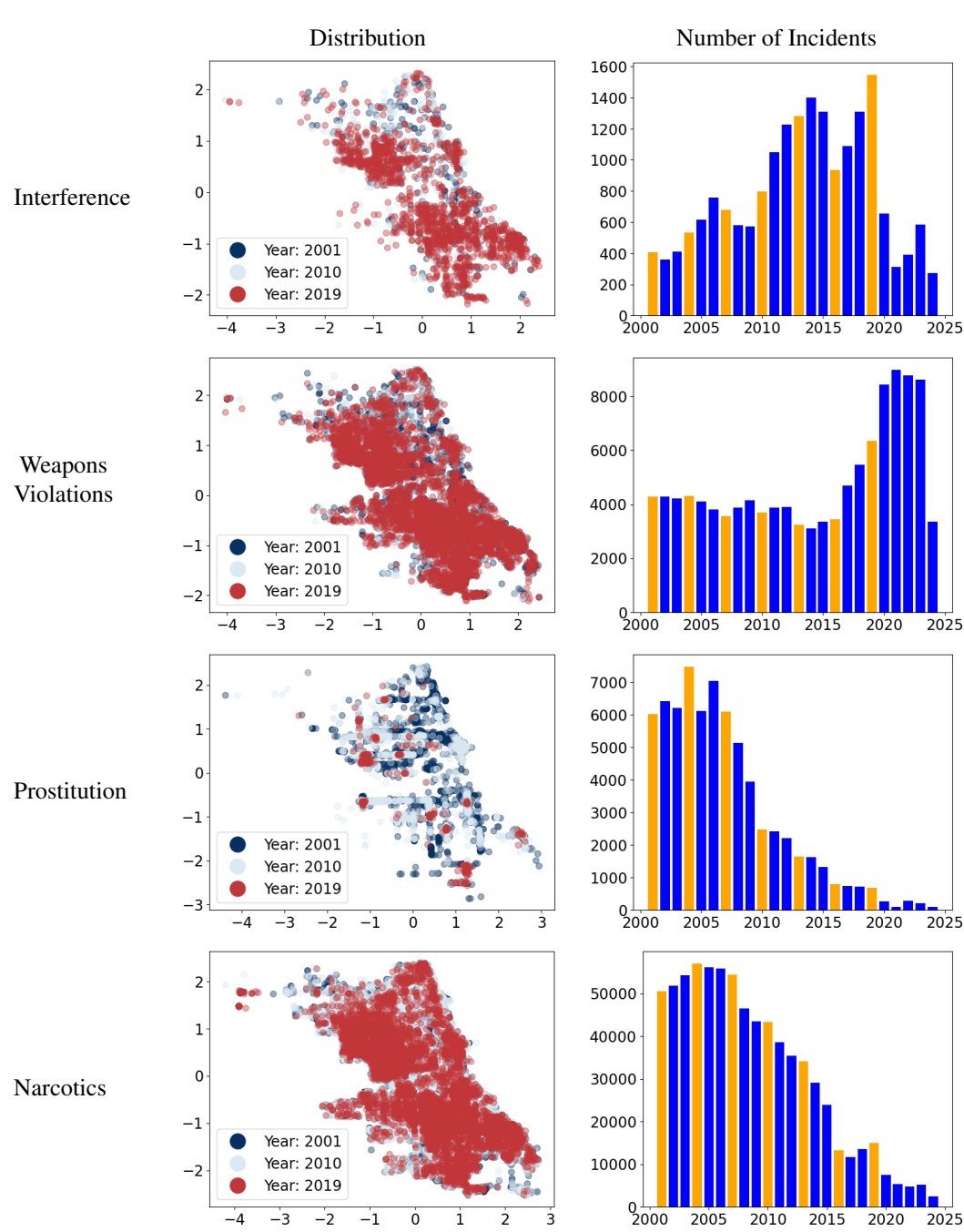

Figure 8: Visualization of crime data in Chicago. (Left) Spatial distribution of incidents in the selected years—early (2001), middle (2010), and late (2019) periods. (Right) Annual number of incidents. The selected years for analysis are highlighted in orange.

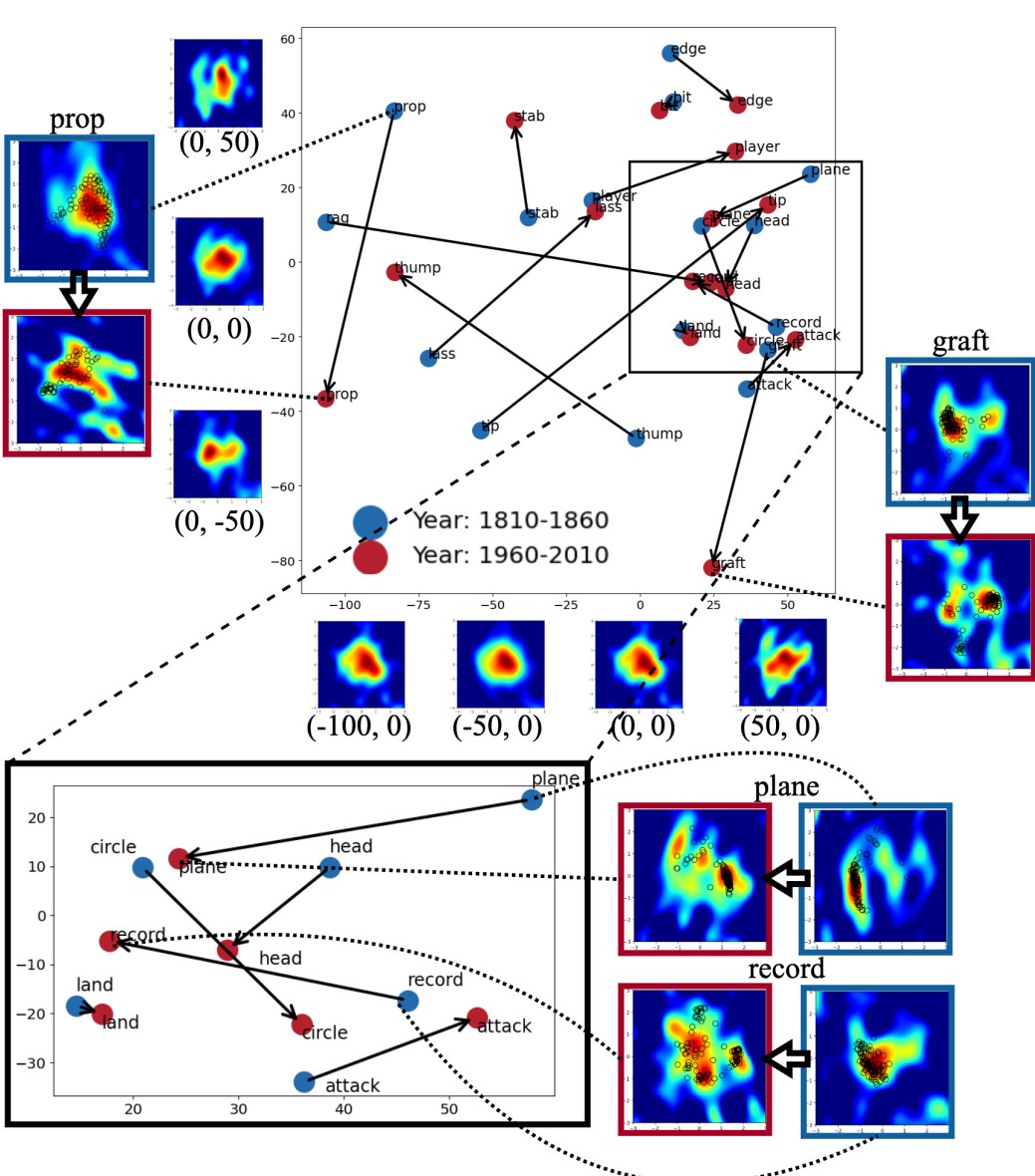

Figure 9: Trajectories of the SemEval dataset in the PC1 and PC2. The central plots show the projection of inferred 30-dimensional weights ($K = 30$ Cosine waves) onto the PC1-PC2 space. To aid interpretation, each principal component axis is annotated with heatmaps that visualize the estimated distributions reconstructed by inverse-mapping a set of evenly spaced points (every 50 units) back to the original RFF space. Surrounding heatmaps represent the estimated distributions over the vector sets (represented by ○) at each time step.

