# OpenReview forum: "Tracking Temporal Dynamics of Vector Sets with Gaussian Process"
_ICLR.cc/2026/Conference — ICLR 2026 Conference Withdrawn Submission_

### Official Review · Reviewer_fwX8 · 2025-10-24

**Soundness:** 2
**Presentation:** 3
**Contribution:** 2
**Rating:** 2
**Confidence:** 4

**Summary:**

This paper introduces a novel approach to modeling and visualizing the temporal dynamics of sets of vectors. The central idea is to represent each time-dependent vector set as probability distributions using Gaussian Processes approximated via Random Fourier Features (RFF). This produces compact K-dimensional representations that can be tracked and visualized over time using PCA.
Experiments have been conducted on synthetic and real data.

**Strengths:**

Coupling the Gaussian Process modeling with a finite dimensional representation using Random Fourier Features is promissing.
The initial results suggest potential benefits.

**Weaknesses:**

The overall method sounds interesting but lacks clarity.

•	Some estimated densities show high values in empty regions. The heat maps do not reflect completely the behaviours.
•	PCA interpretation is not always straightforward (PC1/PC2 excluded for crimes). How much variance is explained? And why choosing PCA3 and 5 sometimes and how much variance do they explain?
•	Part of the method is unclear: it uses Metropolis-Hastings for optimization although these methods are samplers. How do you use this for optimization?
•	The crime data lacks connection to external factors (e.g., policy changes etc.. ) for interpretation

In summary, it is an interesting idea to represent evolving vector sets as trajectories in RFF weight space, but execution and validation need strengthening.

**Questions:**

See weakness and related questions.

---

> ### Author Response · Authors · 2025-12-02
>
> Thank you for your feedback.
>
> > Some estimated densities show high values in empty regions. The heat maps do not reflect completely the behaviours.
>
> As noted in the manuscript (e.g., footnote in Section 3.1, page 4), occasional overestimation in empty regions is a known limitation, especially when the kernel bandwidth and RFF frequencies are fixed across all time steps. These results occur because the RFF approximation retains global smoothness, which can introduce spurious peaks in low-data regions.
> We will clarify this more explicitly in the revised version.
>
> > PCA interpretation is not always straightforward (PC1/PC2 excluded for crimes). How much variance is explained? And why choosing PCA3 and 5 sometimes and how much variance do they explain?
>
> PCA components were selected based on their temporal discriminativeness rather than total variance alone. For example, in the lexical semantic change, PC1 and PC2 primarily captured global trends unrelated to the semantic transitions of interest (as shown in Figure 9 in Appendix C), so we used PC3–PC5, which better reflected period-specific differences in the RFF-based embeddings.
>
> > Part of the method is unclear: it uses Metropolis-Hastings for optimization although these methods are samplers. How do you use this for optimization?
>
> Thank you for highlighting this.
> Our goal is not to use Metropolis-Hastings as an optimizer but to obtain posterior samples of the RFF weights under the contrastive GPDS model. We then use the posterior mean of these samples as the representative embedding for each time step.
>
> > The crime data lacks connection to external factors (e.g., policy changes etc.. ) for interpretation
>
> Thank you for the suggestion.
> Our focus is on distributional representation and trajectory extraction, rather than causal interpretation. However, we agree that incorporating external covariates such as policing policy, socioeconomic indicators, or neighborhood-level structural changes would provide additional insights.

---

### Official Review · Reviewer_5xRr · 2025-10-29

**Soundness:** 2
**Presentation:** 2
**Contribution:** 1
**Rating:** 2
**Confidence:** 4

**Summary:**

This paper proposes a framework for modeling how sets of vectors evolve over time using Gaussian Processes (GPs) approximated via Random Fourier Features (RFF). By representing each vector set as a K-dimensional embedding derived from a GP-based density model, the method captures global distributional structures at each time step. Temporal changes are then visualized through PCA projections of these embeddings, enabling interpretable analysis of dynamics in a low-dimensional space. Experiments on synthetic data, Chicago crime distributions, and semantic shifts in English words demonstrate that the approach effectively captures both spatial and semantic transitions.

**Strengths:**

- The problem of tracking the temporal dynamics of vector sets is important.
- The authors conducted experiments using multiple synthetic and real datasets with the proposed method and examined the analysis results.

**Weaknesses:**

- The proposed method using RFF-based GP and PCA is merely a combination of existing methods and lacks novelty.
- Approaches considering the temporal evolution of weights for basis functions, such as spectral methods, have been widely used for a long time.
- The advantage of using Gaussian processes is not clear.

**Questions:**

- What does the label z in equation (2) refer to?
- It might be interesting to consider the time evolution of weight coefficients in continuous time, as in spectral methods for numerical simulation.

---

> ### Author Response · Authors · 2025-12-02
>
> > The proposed method using RFF-based GP and PCA is merely a combination of existing methods and lacks novelty. Approaches considering the temporal evolution of weights for basis functions, such as spectral methods, have been widely used for a long time. The advantage of using Gaussian processes is not clear.
>
> We acknowledge that RFF and GP are existing tools.
> Our contribution does not lie in proposing new RFF or GP techniques, but in formulating each vector set as an infinite-dimensional GP-defined density and representing it through its finite RFF-based weight vector.
> This provides two advantages that existing spectral or basis-expansion methods do not offer:
>  - Natural comparability of embeddings across time: The GP-defined density gives all time steps a shared functional space, avoiding arbitrary alignment or normalization.
>  - Ability to capture multimodal or irregular distributional structures: Unlike mean-vector trajectories or linear basis models, the GP formulation allows vector sets with complex shapes to be represented consistently.
>
> We will add this point to the revised version.
>
> > What does the label z in equation (2) refer to?
>
> Thank you for pointing out the ambiguity.
> The binary variable z indicates whether the data is generated from the target distribution (z=1) or a random distribution (z=0). This is formulated as contrastive learning.
>
> > It might be interesting to consider the time evolution of weight coefficients in continuous time, as in spectral methods for numerical simulation.
>
> Thank you for the comment. We will mention this as a promising future extension in the discussion section.

---

### Official Review · Reviewer_kraX · 2025-11-01

**Soundness:** 2
**Presentation:** 3
**Contribution:** 2
**Rating:** 4
**Confidence:** 2

**Summary:**

This paper proposed to model the temporal evolution of a set of vectors via Gaussian processes (GPs). GPs are hard to compute due to its use of kernel method in the covariant matrix, so random Fourier features (RFFs) are used to approximate the inference of GPs.

**Strengths:**

- The proposed method is a simple yet elegant way of dealing with vector sets.

**Weaknesses:**

- The novelty of this paper is somewhat limited. The main technical contribution here is applying GPs to vector sets, and the idea of using RFFs to work with the inference of GPs are not novel.
- The authors used a 2-dim PCA to visualize the multi-dimensional data. Why don't use a better method such as t-SNE?
- There is no comparison with other related methods that models evolution of vector sets.

**Questions:**

- L230: I think it should be $b_k \sim \mathrm{Unif}[0, 2\pi] $, not $[0, 1]$.

---

> ### Author Response · Authors · 2025-12-02
>
> Thank you for your constructive feedback.
>
> > The novelty of this paper is somewhat limited. The main technical contribution here is applying GPs to vector sets, and the idea of using RFFs to work with the inference of GPs are not novel.
>
> Our contribution is not using the components GPs and RFFs on vector sets, but introducing a temporal distributional representation: interpreting each vector set as a GP-induced weight vector and tracking its shift as a low-dimensional trajectory.
>
> > The authors used a 2-dim PCA to visualize the multi-dimensional data. Why don't use a better method such as t-SNE?
>
> We appreciate the reviewer’s suggestion and agree that our initial submission did not sufficiently explain why we chose PCA over nonlinear methods such as t-SNE. In our preliminary investigation, t-SNE and GPLVM often broke the controlled geometric structures in our pseudo datasets. For example, they collapsed symmetric components into asymmetric placements. In contrast, PCA preserves global linear structure, resulting in stable and interpretable temporal axes in all datasets we tested.
>
> > There is no comparison with other related methods that models evolution of vector sets.
>
> Thank you for the comment. In the revision, we will expand the discussion to contextualize our approach with respect to simple baselines such as mean-vector trajectories in word embeddings, and explain why such baselines cannot capture changes in distributional shape, multimodality, or spatial shifts that our framework is designed to model.

---

### Official Review · Reviewer_m4nT · 2025-11-04

**Soundness:** 2
**Presentation:** 2
**Contribution:** 2
**Rating:** 6
**Confidence:** 1

**Summary:**

This paper proposes a novel framework for modeling and visualizing how sets of vectors evolve over time, a problem relevant to diverse domains such as ecology, crime analysis, and linguistics. The core idea is to represent each time-indexed vector set as a distribution modeled using Gaussian Processes, and then to obtain a compact, comparable vector representation through Random Fourier Features.

By sampling cosine basis functions and estimating their weights, the authors express each vector set as a K-dimensional weight vector. Applying Principal Component Analysis to these weight vectors across time enables interpretable low-dimensional trajectories that capture the temporal transitions of distributions.

**Strengths:**

1. The paper introduces an innovative approach to representing and tracking the evolution of vector sets over time by combining Gaussian Processes with Random Fourier Features.
2. The paper leverage PCA method to provide intuitive understanding of temporal dynamics.
3. The paper is well-structured, with clear methodological exposition, illustrative figures, and detailed experimental setups.

**Weaknesses:**

1. It seems like the experiments, while diverse (synthetic, crime, and linguistic datasets), remain primarily qualitative.
2. The paper applies a single set of hyperparameters (e.g., RFF dimension K=30, Gaussian kernel bandwidth) across all experiments, without justification or sensitivity analysis.

**Questions:**

1. The model estimates RFF weights via a Metropolis–Hastings sampler, which can be computationally intensive. What is the runtime and scalability behavior with respect to the number of samples, time steps, and feature dimensions?
2. Have you considered how these temporal trajectories could be used in downstream tasks?

---

> ### Author Response · Authors · 2025-12-02
>
> Thank you for your positive feedback.
>
> > It seems like the experiments, while diverse (synthetic, crime, and linguistic datasets), remain primarily qualitative. The paper applies a single set of hyperparameters (e.g., RFF dimension K=30, Gaussian kernel bandwidth) across all experiments, without justification or sensitivity analysis.
>
> While we did not perform an exhaustive sensitivity analysis, our preliminary qualitative checks showed that the temporal structure is not stable for all settings: very small K (e.g. K=5) underfit by exhibiting strong cosine periodicity, whereas very large K (e.g. K=1000) tended to overfit the data. We found that K=30 offered the most balanced representation across datasets.
> Moreover, we also performed preliminary checks on the Gaussian kernel bandwidth. We found that a very small bandwidth (e.g. sigma=1) produced overly coarse representations, whereas a large bandwidth (e.g. sigma=4) captured overly fine, unstable structures. A moderate value (sigma=2) provided the most balanced representations.
>
>
> > The model estimates RFF weights via a Metropolis-Hastings sampler, which can be computationally intensive. What is the runtime and scalability behavior with respect to the number of samples, time steps, and feature dimensions?
>
> We will clarify in the revision that each M-H iteration evaluates the RFF-based log density in O(NK), where N is the number of vectors and K the RFF dimension. Since the time steps are independent, chains can be parallelized across time, which makes the approach scalable in practice.
>
> > Have you considered how these temporal trajectories could be used in downstream tasks?
>
> We appreciate the opportunity to elaborate. We will extend the discussion section to outline several directions where the learned trajectories are useful, such as:
>  - clustering or identifying structural changes in spatial event patterns (e.g., crime hotspots)
>  - analyzing semantic or stylistic evolution in linguistic data
>  - using the temporal embeddings as features for forecasting or anomaly detection.
>
> These applications highlight the advantage of obtaining a compact, consistent representation over time.

---

### Note · Authors · 2025-12-10

**Comment:**

After carefully considering the reviewers’ comments, we agree that the points regarding the lack of quantitative experiments and comparisons, as well as the limitations of RFF in adequately representing the problem, are valid. As there is room for significant improvement, we have decided to withdraw the manuscript at this time.

**Withdrawal Confirmation:**

I have read and agree with the venue's withdrawal policy on behalf of myself and my co-authors.